# CD103^+^ cDC1 Dendritic Cell Vaccine Therapy for Osteosarcoma Lung Metastases

**DOI:** 10.3390/cancers16193251

**Published:** 2024-09-24

**Authors:** Yuanzheng Yang, Yifan Zhou, Jian Wang, You Zhou, Stephanie S. Watowich, Eugenie S. Kleinerman

**Affiliations:** 1Department of Pediatrics Research, The University of Texas MD Anderson Cancer Center, 1515 Holcombe Boulevard, Houston, TX 77030, USA; yyang9@mdanderson.org (Y.Y.); yzhou15@mdanderson.org (Y.Z.); 2Department of Immunology, The University of Texas MD Anderson Cancer Center, 1515 Holcombe Boulevard, Houston, TX 77030, USA; yfan_zhou@smu.edu.cn (Y.Z.); swatowic@mdanderson.org (S.S.W.); 3Department of Laboratory Medicine, Zhujiang Hospital, Southern Medical University, Guangzhou 510282, China; 4Department of Biostatistics, The University of Texas MD Anderson Cancer Center, 1515 Holcombe Boulevard, Houston, TX 77030, USA; jianwang@mdanderson.org; 5Department of Cancer Biology, The University of Texas MD Anderson Cancer Center, 1515 Holcombe Boulevard, Houston, TX 77030, USA

**Keywords:** CD103^+^ cDC1 dendritic cells, vaccine therapy, anti-CTLA-4, anti-tumor efficacy, osteosarcoma, lung metastases

## Abstract

**Simple Summary:**

Patients with relapsed, refractory, or metastatic osteosarcoma have few therapeutic options. Our findings demonstrated the efficacy of a DC vaccine generated using CD103^+^ cDC1 cells and OS cell lysates against the primary tumor and lung metastases, as well as its ability to induce a systemic immune response. This cDCV therapy elicited an increase in the infiltration of CD8+ T-cells into the primary and metastatic tumors as well as the TdLNs. cDCV efficacy was increased when combined with anti-CTLA-4. The failure of immunotherapies has been linked to their inability to induce infiltration of T-cells into the tumor. Therefore, our data suggest the potential for this novel immunotherapy using innate immune cells (type 1 CD103^+^ dendritic cells) alone or in combination with checkpoint blockade for patients with relapsed or metastatic OS, a tumor type that is refractory to the majority of current immunotherapies.

**Abstract:**

Background: We generated a CD103^+^DC vaccine using K7M3 OS cell lysates (cDCV) and investigated its ability to induce regression of primary tumors, established lung metastases, and a systemic immune response. Methods: A bilateral tumor model was used to assess cDCV therapy efficacy and systemic immunity induction. K7M3 cells were injected into mice bilaterally. Right-sided tumors received PBS (control) or cDCV. Left-sided tumors were untreated. Tumor growth was compared between the *vaccine-treated* and *untreated* tumor on the contralateral side and compared to the control group. The immune cell profiles of the tumors, and tumor-draining lymph nodes (TdLNs) and spleen were evaluated. To determine the efficacy of *systemic* cDCV therapy against *established* lung metastases, K7M3 cells were injected intratibially. Leg amputation was performed 5 weeks later. Mice were treated intravenously with PBS or cDCV and euthanized 6 weeks later. Lungs, TdLNs and spleen were collected. The number and size of the lung nodules were quantified. The immune cell profile of tumor, and lymph nodes and spleen were also evaluated. Using this same model, we evaluated the effect of cDCV + anti-CTLA-4. Results: cDCV therapy inhibited the treated and untreated tumors and increased the number of T-cells in these tumors and the lymph nodes compared to control-treated mice. Systemic cDCV therapy administered following amputation decreased the size and number of lung metastases, and increased T-cell numbers in the tumor and lymph nodes. Combining anti-CTLA-4 with cDCV therapy increased cDCV efficacy against lung metastases. Conclusions: *Intratumor* cDCV generated a systemic immune response inhibiting the growth of both the treated and untreated tumors, with increased T-cells in the tumor and lymph nodes. *Systemic* cDCV was effective against established lung metastases. Efficacy was increased by anti-CTLA4. cDCVs may provide a novel therapeutic approach for relapsed/metastatic OS patients.

## 1. Introduction

The cure rates for osteosarcoma (OS) using multiagent chemotherapy and surgery have not improved for over 25 years [1,2,3,4,5]. Dose intensification for newly diagnosed patients who respond poorly to neoadjuvant chemotherapy failed to improve patient outcomes [6]. The overall cure rates of 60–65% leave approximately 40% of patients who relapse after initial therapy. These patients, along with those that present with metastases at diagnosis, have an extremely poor prognosis with a long-term survival rate of under 20%. Salvage chemotherapy for these patients has shown limited efficacy and no new therapies were identified in seven recent phase II trials from the Children’s Oncology Group [7]. The activity of checkpoint inhibitor immunotherapy which targets T-cell activation has been disappointing in sarcoma patients [8]. However, the demonstrated therapeutic activity of L-MTP-PE (a macrophage activating agent) in relapsed and newly diagnosed OS patients [9,10] suggests that other immunotherapies targeting “innate” immune cells should be explored particularly for patients with relapsed/metastatic disease. The success of L-MTP-PE therapy in improving overall survival in newly diagnosed patients and in patients with OS lung metastases also suggests that OS patients can mount an immune response, which further support a cDCV-based therapeutic approach.

Dendritic cells (DCs) are innate immune antigen-presenting cells that link innate and adaptive immunity, present antigen to T-cells, and are pivotal in generating adaptive immunity. Type 1 conventional DCs (cDC1s) are critical in the T-cell-mediated anti-tumor response as well as being essential in the antiviral immune response in the lung. Using DCs to generate cancer vaccines is an emerging immunotherapy focus. DC vaccines derived from monocytes (MoDCVs) have been shown to be well tolerated with no significant toxicity in children, adolescents, and young adults with relapsed OS [11]. While safety was established and no maximum tolerated dose achieved, therapeutic activity and the induction of an immune T-cell response were seen in only 3 of 12 patients. Immunotherapies are typically most effective against minimal residual disease, as was the case for L-MTP-PE [9,10], which may explain the low anti-tumor efficacy in this trial. However, patients with refractory bone and soft tissue sarcomas with metastases in the shoulder, spine, pelvis, and lung showed sclerotic changes in these lesions after MoDCV (a sign of response in sarcoma) with stable disease for 5 months [12]. Additional support for the potential of DC vaccine therapy for relapsed/metastatic sarcoma patients is provided by a Phase I trial where 18 sarcoma patients received activated DCs intratumorally. Eleven of the eighteen patients were alive and disease-free at 2–8 years [13].

We developed a unique DC vaccine using in vitro-cultured CD103⁺DCs (cDCVs). When injected *intratumorally*, this novel cDCV was superior to the MoDCV in inducing an anti-tumor immune response and in activating a *systemic* T-cell immune response [14,15]. Using an *experimental* OS metastasis model, we demonstrated that a cDCV generated using OS tumor cell lysates injected 30 days *prior* to the intravenous injection of OS cells significantly reduced the formation of lung metastases with increased infiltration of IFNɣ^+^CD8⁺ T-cells into tumors, spleen, and tumor-draining lymph nodes (TdLNs) [14]. These data show that the cDCV was effective in *preventing* metastatic development [14]. However, these studies did not evaluate cDCV efficacy against OS tumors that had already metastasized to the lung. Since our novel cDCV was shown to be superior to a MoDCV in anti-tumor activity, particularly in inducing T-cell activation and the infiltration of tumor antigen-specific CD8⁺ T-cells [14], our prior findings support the concept of investigating the potential of a DC vaccine derived from CD103⁺cDC1 cells for activity against *established* OS lung metastases, which is the area of need in OS as 90% of newly diagnosed patients have micro-metastases in the lung at the time of diagnosis.

An orthotopic model where OS cells are injected into the bone and spread to the lung from the primary tumor is the optimal model to investigate therapeutic efficacy against established lung metastases as this model is more reflective of the clinical disease. In this model, OS cells spread from the bone to the lung rather than being injected intravenously. Using this model, the studies described herein show that cDCV therapy injected systemically following *removal of the primary tumor* at a time when lung metastases are present induced regression of the established lung metastases and the infiltration of T-cells into the lung tumor and TdLN. We also demonstrated that this anti-tumor activity was enhanced by anti-CTLA-4. Our studies indicate that cDCV therapy may provide a novel therapeutic approach for relapsed OS patients with metastases in the lung, thereby improving survival for this difficult patient population with few alternative therapies.

## 2. Materials and Methods

### 2.1. Mice, Cell Line, and Reagents

Balb/c mice were purchased from Charles River (Wilmington, MA, USA) and maintained at the University of Texas MD Anderson Cancer Center in a specific pathogen-free animal facility approved by the American Association for Accreditation of Laboratory Animal Care. Age-matched mice were chosen randomly for treatment groups and were used in accordance with MD Anderson Cancer Center’s Institutional Animal Care and Use Committee-approved protocol (No. 00000896).

The mouse metastatic K7M3 cell line was derived from K7M2 cells in our laboratory [16] and cultured for no more than 30 passages. The unique signature identification of the K7M3 cells was confirmed by a short-tandem-repeat DNA microsatellite fingerprinting analysis carried out by IDEXX BioAnalytics (Columbia, MO, USA). The cell culture was tested for mycoplasma species every other month using the MycoAlert Mycoplasma Detection Kit (Lonza Bioscience, Houston, TX, USA) and was confirmed to be negative.

CTLA-4 antibody (clone 9H10) was purchased from Bio X-Cell company (West Lebanon, NH, USA) and injected intraperitoneally (i.p.) (200 µg/mouse for the first treatment, 100 µg/mouse for subsequent treatments) twice a week to the mice one week after DCV treatment for a total of four weeks of treatment.

### 2.2. Isolation of CD103^+^ cDC1 Cells

Ex vivo generation and expansion of CD103^+^ cDC1s cells from Balb/c mice bone marrow was based on our previous studies [14] and optimized for maximal production efficiency and high yields of CD103^+^ cDC1s cells. Bone marrow cells were collected from the femurs of 6-to-8-week-old Balb/c mice and cultured in RPMI 1640 complete medium supplemented with 50 µM β-mercaptoethanol, 50 ng/mL human FMS-like tyrosine kinase 3 ligand (Flt-3L, PeproTech, Rocky Hill, NJ, USA) plus 0.5 ng/mL murine granulocyte-macrophage colony-stimulating factor (GM-CSF, PeproTech, Rocky Hill, NJ, USA) at 1.5 × 10^6^ cells/mL. Fresh full medium was supplemented on day 5 and day 7; suspension cells were collected at day 9 and cultured in fresh medium supplemented with 50 µM β-mercaptoethanol, 50 ng/mL Flt-3L plus 2 ng/mL murine GM-CSF at 0.5 × 10^6^ cells/mL, and fresh full medium was supplemented at day 13 once to maintain the culture. Non-adherent cells were collected at 17 days (Appendix A) and stained with CD11b, CD11c, CD103, CD172α, and CD24 antibodies together with GhostDye Violet 510 (Tonbo Biosciences, San Diego, CA, USA). CD103^+^ cDC1s (CD11c^+^, CD24^+^, CD172α^-^, and CD103^+^ cells) populations were sorted using a FACSAria Fusion Cell Sorter (BD Biosciences, Palo Alto, CA, USA) (Appendix A).

### 2.3. Generation of the CD103^+^ cDC1 Vaccine (cDCV) Using K7M3 Tumor Lysates

Log phase cultured K7M3 cells were washed and resuspended in PBS at 50 × 10^6^ cells/mL. The cell suspension was frozen and thawed ×3 and then ultrasonically disrupted for 50 s three times on ice. Complete disruption of the cells was confirmed by trypan blue staining. CD103^+^ cDC1s were incubated with K7M3 tumor cell lysate and polyinosinic–polycytidylic acid (poly-I:C) to stimulate cDCV maturation and antigen loading using a ratio of two tumor cells to one CD103^+^ cDC1 cell and incubated for 4 h at 37 °C. These poly-I:C-stimulated and antigen-loaded CD103^+^ cDC1 vaccine cells (cDCV) were washed twice with phosphate-buffered saline (PBS) to eliminate the tumor lysate debris and resuspended in sterile PBS. Therapy consisted of 2 × 10^6^ cDCV cells per treatment.

### 2.4. cDCV Therapy for Primary and Metastatic Osteosarcoma

A quantity of 2 × 10^6^ K7M3 cells was subcutaneously (s.c.) injected on both sides of the mice’s abdomens. Tumors on the right were treated intratumorally (i.t.) with cDCV or PBS on days 4 and 7 following implantation, a time when the tumors were palpable. The tumor size (length × width) was measured every 3–4 days using a caliper. Mice were euthanized 5 weeks after therapy.

We used our intra-bone injection OS model [16] to evaluate therapy activity of cDCV against spontaneous lung metastasis. Mice were injected with 0.25 × 10^6^ K7M3 cells into the right tibia. The tumors were verified by X-ray imaging at 5 weeks and a leg amputation was performed 24 h later. This prevents further seeding of tumor cells from the leg to the lungs. Three days after amputation the mice were injected intravenously (i.v.) with 2 × 10^6^ cDCV cells twice in the week. Mice were euthanized 5 weeks later. Lungs, spleens, and tumor draining lymph nodes (TdLNs) were collected for analysis.

### 2.5. Immune Profiling of TdLNs and Spleens by Flow Cytometry Analysis

Single-cell suspensions were generated from spleens and inguinal TdLNs, which were closing to the subcutaneous tumor formation as previously described [14]. The inguinal lymph nodes were also analyzed for immune cell infiltration in the orthotopic investigation where the primary tumor was in the tibia. Cell surface Fc receptors were blocked with rat anti-mouse CD16/32 (Tonbo Biosciences) for 15 min at 4 °C. For the T-cell subset analysis, T-cells were stimulated with 0.5 μg/mL ionomycin (Sigma, St Louis, MI, USA) and 50 ng/mL phorbol 12-myristate 13 acetate (PMA) (Sigma, St Louis, MI, USA) for 4 h in the presence of GolgiStop (BD Biosciences, Palo Alto, CA, USA). For surface maker staining, isolated cells were incubated with fluorescent dye-conjugated antibodies for 30 min at 4 °C; for intracellular protein staining, T-cells were fixed and permeabilized using Intracellular Fixation and Permeabilization Buffer (eBiosciences, San Diego, CA, USA) first, and stained with antibodies against intracellular proteins for 30 min at 4 °C. The following antibodies were used: PE-Cy7-conjugated CD11c (N418) or MHC-II (M5/114.15.2); PE-conjugated CD103 (2E7) or IFN-γ (XMG 1.2) antibodies; APC-Cy7-conjugated CD4 (GK1.5) or CD45 (30-F11) antibodies; PerCP-Cy5.5-conjugated Foxp3 (FJK-16S), CD19 (1D3), or CD11b (M1/70) antibodies; FITC-conjugated CD172α (P84), MHC-I (AF6-88.5), or Ly6G (1A8) antibodies; BV605-conjugated CD3e(17A2) antibody; BV711-conjugated CD8α (53–6.7) or F4/80 (BM8) antibodies; BV421-conjugated IL-4 (11B11) or XCR1 (ZET) antibodies. All antibodies were purchased from BioLegend (San Diego, CA, USA), eBioscience (San Diego, CA, USA), BD Biosciences, or Tonbo Biosciences. Dead cells were eliminated in all experiments using Ghost DyeTM Violet 510 (Tonbo Biosciences). An LSRFortessa flow cytometer (BD) was used to perform the flow cytometry analysis. Gating was performed as previously described [14]. Data analysis was performed on viable single cells (Ghost Dye) using FlowJo software (version 10, TreeStar, Ashland, OR, USA).

### 2.6. Immunofluorescence Staining

Slides of frozen tumor sections (5 µm) were fixed in cold Acetone for 5 min then cold Acetone + Chloroform (1:1) for 5 min; followed by cold Acetone for 5 min, and finally permeabilized by 0.25% TritonX-100 in PBS for 5 min. After three PBS washes, the sections were blocked by 5% normal goat serum plus 1% normal horse serum in PBS for 30 min and by 4% fish gel in PBS for another 30 min at room temperature. In a humid chamber box, sections were incubated with anti-CD3-AF488 (1:200), CD8α-AF594 (1:200), CD4-AF647 (1:100), or anti-CD103-AF488(1:100), and CD11c-AF594 (1:200) at 4 °C overnight, then washed ×3 with PBS; the slides were then counterstained with Fluoro-Gel II with DAPI and mounted with a glass cover slip. Three tumor frozen samples from each treatment group were sectioned for CD3/CD4/CD8 staining and CD11c/CD103 staining. Tumor areas from each section were imaged at 400× by Leica DMi8 microscope (Leica Mi-crosystems, Wetzlar, Germany). Positive staining was quantified in eight random microscopy fields from three tumor samples using Simple PCI software (version 6.6, Hamamatsu, Sewickley, PA, USA), and the average expression was calculated as average ± SEM.

Primary antibodies anti-CD86 (1:200) and anti-CD163 (1:200) (CD86, NeoBiotechnologies, Union City, CA, USA; CD163, Abcam, Waltham, MA, USA) were used for the macrophage markers. After stained at 4 °C overnight and washed ×3 with PBS, the sections were incubated with the secondary antibody (1:500, Alexa Fluor 488 goat anti-rabbit, Invitrogen, Waltham, MA, USA) at room temperature for 1 h in the dark and washed ×3 with PBS. Slides were then counterstained with Fluoro-Gel II with DAPI and mounted as above. Tumor areas from each section were imaged at 400× by Leica DMi8 microscope (Leica Microsystems, Wetzlar, Germany). Positive staining was quantified in eight random microscopy fields from three tumor samples using Simple PCI software (Hamamatsu, Sewickley, PA, USA), and the average expression was calculated as average ± SEM.

### 2.7. Statistical Analysis

All values reported represent means ± SEMs. A 2-tailed Student *t* test was used to determine significance. For the mouse experiments, the statistical significance was evaluated using the Mann–Whitney U-test and the one-way ANOVA by GraphPad Prism 9 (GraphStats, Bangalore, India). For the tumor growth experiments, a linear mixed effects model was used to compare the tumor growth rates among different groups. The tumor size was log-transformed to ensure normally distributed residuals and homogeneity of variance over time in the model. The analysis was performed using R package (R Development Core Team). *p* values less than 0.05 were considered statistically significant.

## 3. Results

### 3.1. cDCV Therapy Induces Systemic Tumor Immunity against Primary OS

The bilateral tumor model was used to determine whether intratumor injection of the cDCV generated using OS tumor cell lysates induced systemic tumor immunity against an established *untreated* tumor. K7M3 cells were injected subcutaneously on the right and left side of the abdomen (Figure 1A). Treatment began 4 days after tumor cell injection. Right-side tumors were injected intratumorally with PBS or the cDCV. The left-side tumors were left untreated. Tumors were measured every 3–4 days for 6 weeks. The cDCV-treated mice showed a significant decrease in tumor growth in both the treated and *untreated* contralateral tumors compared to the mice treated with PBS control (Figure 1B, Appendix A). There was no significant difference between the cDCV-treated and untreated tumors in the cDCV-treated mice. The data presented here demonstrate that intratumoral injection of cDCV induced systemic tumor immunity in addition to a direct anti-tumor response.

Since T-cell activation and infiltration plays a critical role in the efficacy of cDCV therapy, we analyzed the bilateral treated and untreated tumors collected after cDCV therapy for immune cell infiltration and the immune cell profile comparing these with the immune cell profile in the control tumors. Both the treated and contralateral untreated tumors from the cDCV-treated mice showed a significant increase in CD3^+^, CD8^+^, and CD4^+^ T-cells (Figure 2A–C,K,L). However, although there was an increase in the total number of CD3^+^, CD8^+^, and CD4^+^ cells, cDCV treatment did not alter the ratio of CD8^+^/CD3^+^ and CD4^+^/CD3^+^ T-cells in either the left or the right-sided tumors when compared with the PBS-treated mice (Figure 2D,E).

There was also no difference in the number of CD86^+^ (M1) or CD163^+^ (M2) macrophages (Figure 2F,G) in the DCV-treated and untreated tumors. Since activated T-cells were increased in both the treated and untreated tumors, these data support the conclusion of induction of a systemic response to intratumor DCV therapy.

Immunohistochemical staining showed that there was no significant difference in total number of CD103^+^ or CD11c^+^ cells between control and cDCV-treated tumors or between the treated and untreated tumors. There was also no difference in the ratio of CD103^+^/CD11c^+^ cells (Figure 2H–J). We previously showed that only a fraction of cDCs from the initial vaccine could be detected 40 h after intratumor delivery, consistent with DCs having a short lifespan (2–3 days) in vivo [14]. Since our analysis of DC infiltration was 5 weeks after cDCV injection, the lack of an increase in DCs was expected.

### 3.2. cDCV Reduces Established OS Lung Metastases

We previously showed that cDCV therapy given 35 days *prior* to the *intravenous* injection of OS cells *prevented* the formation of lung metastases. This does not mimic the clinical situation where lung metastases are already present. To determine whether systemic DCV therapy was effective against *established* lung metastases that metastasized from the *primary tumor*, K7M3 cells were injected into the right tibia (Figure 3A). Primary tumor formation was confirmed at 5 weeks post-injection by X-ray imaging as previously described ([17]; Figure 3B). Mice were sacrificed 1 week later, and lungs were extracted and analyzed for the presence of lung metastases at this timepoint. Macroscopic lung metastases were confirmed by fixing the lung with Bouin’s solution for 24 h (Figure 3C). Hematoxylin and eosin staining of lung sections also confirmed the presence of microscopic metastases at this time point (Figure 3D).

Having shown that lung metastases were present 5 weeks after tumor cell injection into the tibia, we next performed a leg amputation to prevent further seeding of the tumor cells from the primary tumor to the lungs. Intravenous cDCV therapy was given twice starting 3 days after amputation (Figure 3A). Six weeks after therapy completion, lungs were collected. There was a significant decrease in the number and size of lung metastases in the DCV-treated mice (Figure 3E,F). Moreover, three of seven mice had *no evidence of disease* after the cDCV treatment suggesting complete eradication of the established tumors. By contrast, all of the control mice had lung tumor nodules, and these nodules were significantly larger than those in the cDCV-treated mice (Figure 3F).

### 3.3. Effect of Combining cDCV Therapy with Anti-CTLA-4 Therapy

We previously showed that combining anti-CTLA-4 with cDCV therapy increased cDCV efficacy against a subcutaneous tumor over and above that seen with either anti-CTLA-4 or cDCV alone [14]. We therefore investigated the efficacy of combination therapy against *established* lung metastases by repeating the orthotopic studies described above. K7M3 cells were injected into the right tibia, and the primary tumors were allowed to grow for 5 weeks. Following amputation, mice were treated with PBS (control), anti-CTLA-4, cDCV, or anti-CTLA-4 plus the cDCV (Figure 4A). Lungs, TdLNs, and spleens were collected one week after the last anti-CTLA-4 treatment (6 weeks after amputation) and analyzed as described above. There was a significant decrease in the number of lung tumor nodules in the mice treated with cDCV or anti-CTLA-4 alone (Figure 4B). Therapy efficacy was significantly increased in mice treated with combination therapy compared to either cDCV or anti-CTLA-4 therapy alone (*p* < 0.05).

An analysis of the T-cells in the tumor nodules showed a significant increase in CD8^+^ cells in the tumors from cDCV-treated mice compared to tumors from control mice (Figure 5A), while anti-CTLA-4 treatment alone had no effect on CD8^+^ cell infiltration in the tumor. Combination therapy of cDCV with anti-CTLA-4 also showed a significant increase in CD8^+^ cells in the tumors compared to control, but there was no significant difference in CD8^+^ cells between the cDCV and combination therapy. There was no difference in the number of intratumor CD4^+^ cells in any of the treatment groups (Figure 5B).

Immunofluorescence staining of CD8^+^ T-cells in the lung tumors from control mice showed T-cells around the periphery of the tumor with few inside the tumor (Figure 5C). Tumors from mice treated with anti-CTLA-4 alone also showed poor T-cell penetration (Figure 5D). By contrast, tumors from the cDCV-treated mice showed T-cell infiltration into the tumors (Figure 5E). T-cell infiltration was also seen in tumors from mice treated with combination therapy (Figure 5F).

We also evaluated the immune cell profile of the spleens and inguinal lymph nodes (which are the TdLNs for the primary bone tumor). CD8^+^ T-cells, CD4^+^ T-cells, and cytotoxic T-cells (TC1) were significantly increased in the lymph nodes from the cDCV-treated mice (Figure 6A–C). By contrast there was no statistical difference in CD8^+^ T-cells, CD4^+^ T-cells, and TC1 cells in response to anti-CTLA-4 therapy or combined therapy. There was no statistical difference in the number of Th1, T-reg, or Th17 cells in the lymph nodes among control and single treatment or combination treatment (Figure 6D–F). DCs were significantly increased in lymph nodes *only* after cDCV therapy (Figure 6G), while other myeloid subsets were not statistically different in any of the treatment groups (Figure 6H–J). There was no difference in the T-cells or myeloid subsets in the spleens from the control, cDCV, anti-CTLA-4, or combination therapy-treated mice (Appendix A). These data indicate that unlike the lymph nodes, the spleen does not reflect the T-cell infiltration seen in the metastatic lung tumors.

## 4. Discussion

In summary, we demonstrated that a DC vaccine generated using the CD103⁺cDC1 subset and loaded with OS cell lysates as a source of tumor antigens induced tumor regression when injected into the primary tumor and stimulated a systemic immune response resulting in the regression of the untreated tumor on the contralateral side. The activity of the cDCV was accompanied by an increased infiltration of CD8⁺ and CD4⁺ cells into both the treated and untreated tumors. The finding of increased infiltration of CD8⁺ and CD4⁺ cells in the untreated contralateral tumor supports the generation of a systemic response. By contrast, there was no change in the number of intratumor M1 or M2 macrophages. While the total number of CD4⁺ and CD8⁺ T-cells increased, their percentage to the total T-cell number (CD3⁺ cells) did not change. The findings of no change in M1 and M2 macrophages is important, as we previously showed that the activity of anti-PD1 therapy against OS lung metastases was mediated by the increased infiltration of M1 macrophages into the tumor and a decrease in M2 macrophages [18]. Furthermore, M2 macrophages suppress the immune response and are a negative prognostic factor for OS [19,20,21]. Taken together, these data suggest that the anti-tumor activity of the cDCV therapy was mediated by the activation and infiltration of cytotoxic T-cells.

While we did not demonstrate complete tumor eradication, this is not surprising, as only 2 cDCV injections were given and mice were *untreated* for ~5 weeks. Curative therapy will most likely require multiple cDCV injections for longer periods of time, similar to what was used clinically with other immunotherapies targeting *innate* immune cells, i.e., L-MTP-PE where therapy continued 1–2 times weekly for 6–9 months [9,10,22]. The failure of a pervious MoDCV in relapsed OS patients may be secondary to the short 3-week treatment period [11], in addition to the inferior activity of MoDCVs compared to DCVs generated using CD103^+^ cells [14]. This will be important when translating this therapeutic approach. In addition, we anticipate that improved cDCV therapeutic efficacy will be achieved by initiating therapy in the setting of residual *microscopic* disease, as is the case in newly diagnosed OS patients following primary tumor resection or patients with lung metastases that have been resected. As we showed with L-MTP-PE, we also anticipate that cDCV therapy can be administered together with the chemotherapy agents used to treat OS [10,22,23,24].

Having demonstrated the successful anti-tumor activity of the cDCV following *intratumor* injection with the generation of a systemic response, we evaluated *systemic* vaccine effectiveness against *established* lung metastases that arose from the primary tumor. This was carried out using our orthotopic mouse model, which is more relevant to the patient population where 90% of newly diagnosed patients have microscopic spread to the lung at the time of diagnosis prior to chemotherapy administration, with only a 65% 5-year survival rate following combination chemotherapy. In this model, OS cells are injected into the tibia and a tumor is allowed to grow for 5 weeks, at which time there are positive X-ray findings documenting primary tumor formation and evidence of lung metastases. Thus, this is a model of *established*, as opposed to *experimental* metastases [16], and more importantly represents *spontaneous* metastatic spread from the primary tumor in the leg into the lungs. Amputation of the affected limb prevents any further metastatic spread, allowing for the assessment of systemic therapeutic activity against established metastases. Using this clinically relevant model, we showed that intravenous cDCV therapy decreased the number and size of the lung metastases, with three of seven mice having no evidence of disease. The findings that the sizes of the metastases in the cDCV-treated mice were significantly smaller and that three of seven mice had no evidence of either macro- or microscopic tumors further supports the successful anti-tumor activity of the cDCV therapy. We quantified T-cell numbers in these small tumors. As seen after intratumor cDCV therapy, there was a significant increase in CD8⁺ T-cells in the mice treated with systemic cDCV, which again supports the induction of an immune response and is significant, as cDCV efficacy has been shown to be associated with CD8⁺ T-cell activation and tumor infiltration [14].

The results of our investigations demonstrate that both the intratumor and intravenous route of cDCV delivery are effective in generating an effective anti-tumor response. The intratumor delivery resulted in both a local and systemic response in primary tumors (untreated and treated) while the intravenous cDCV delivery was effective against *established* lung metastases. Since we are targeting established lung metastases, systemic cDCV delivery is required. Intratumor vaccination is not appropriate. This study is the *first* to demonstrate the efficacy of systemic cDCV therapy against *established* OS lung metastases that arise from the primary tumor in the bone. The prior studies evaluated the effect of *intratumor* cDCV administration on the growth of a subcutaneous tumor and whether pretreating mice 1 month *before* tumor cell injection *prevented the development* of lung metastases. These studies did not evaluate the effect of cDCV intratumor therapy on an *untreated* OS tumor on the contralateral side of the mouse, or more importantly its activity against established lung metastases that were present *before* systemic therapy is initiated. The prior studies support the *concept* of using cDCV therapy for relapsed OS but are not sufficient to justify *translation* of cDCV therapy for the treatment of OS lung metastases. Demonstrating that a therapy can *prevent the formation* of metastases in the lung is not therapeutically equivalent to inducing the *regression of metastatic tumors* that are already growing in the lung. The lung tumor microenvironment (TME) has been shown to play a critical role in the successful growth of the OS cells once the cells have metastasized to the lung and to therapy response [25]. Eliminating the circulating OS cells *before* they reach the lung or interfering with the early metastatic process *before* a tumor is formed in the lung does not mean that the therapy will be effective in patients who have OS lung metastases. Therefore, it is imperative that the potential of any new therapy aimed at treating patients with OS lung metastases be evaluated in the context of the lung TME.

As stated above, the majority of newly diagnosed OS patients have *microscopic* disease at the time of diagnosis. Standard three-drug chemotherapy cures ~65% of OS patients. Unfortunately, neither dose intensification nor the addition of additional chemotherapy agents has been successful in increasing this 65% survival rate [6]. The prognosis for patients who relapse with OS lung metastases is dismal (~20% 2-year survival) and has not changed in >25 years. The majority of patients with lung metastases can have these tumors excised putting them in the setting of minimal residual/microscopic disease. However, 85% of patients will relapse again within 1 year. It has been shown that DC vaccine therapy is best for patients with low tumor burden, thereby providing additional justification for its use in this patient population [26]. Patients with bone metastases have an even poorer prognosis. We demonstrated the induction of systemic immunity in addition to activity against lung metastases. Therefore, cDC-based vaccines using antigen-loaded CD103⁺cDC1 cells may be an alternative therapy for patients with disseminated, relapsed, or metastatic OS. The ability to inject one tumor site and induce systemic anti-tumor immunity may make cDCV-based therapy appropriate for OS patients with skip bone lesions and metastases outside the lung. These patients have the worst prognosis with no effective therapies at the present time. Although these cases are rare, the lack of effective therapeutic options necessitates the identification of newer approaches.

We demonstrated an increase in the number of CD8⁺ T-cells in tumors from cDCV-treated mice, which was associated with the positive therapeutic response. Successful cDCV anti-tumor efficacy has been shown to be linked to CD8⁺ T-cell activation and infiltration into the tumor and the TdLNs [11,12,14]. Anti-CTLA-4 is a checkpoint inhibitor that enhances T-cell priming in the lymph nodes and increases T-cell activation and proliferation [27]. We speculated that adding anti-CTLA-4 may improve cDCV efficacy by increasing the number of activated CD8⁺ T-cells in the tumors. Our data indeed demonstrated that combining cDCV with anti-CTLA-4 increased the therapeutic activity against established lung metastases, however there was no significant increase in the number of CD8⁺ cells in the tumors compared to cDCV-treatment alone. This may be due to the fact that tumors from both treatment regimens were small, making it difficult to see significant changes.

Since T-cell penetration into the tumor is critical for immunotherapy success, in addition to quantifying the number of CD8+ cells, we also examined the location of the T-cells. The control tumors showed T-cells primarily located around the periphery of the tumor. By contrast, T-cells were found within the tumor in the cDCV-treated mice. Intratumor T-cells were also seen within the tumors from mice treated with combination therapy but not those treated with anti-CTLA-4 alone. There was no detectable difference in the number of intratumor T-cells in tumors from mice treated with combination therapy versus cDCV alone. It is noteworthy that only tumors from mice that had received cDCV as part of their therapy (cDCV alone or in combination with anti-CTLA-4) showed this pattern of CD8+ T-cell infiltration, suggesting that this effect was mediated by the cDCV. This also suggests that the therapeutic activity of other immunotherapies that depend on T-cell activation and tumor infiltration may be enhanced by combining with a cDC1 vaccine. At the present time, we do not have an explanation for the enhanced activity of the combination therapy compared to cDCV therapy alone in terms of T-cell numbers or penetration into the tumor. Our analysis was performed 6 weeks after therapy completion. Earlier analysis may reveal differences between the combination and single therapies that can explain the therapeutic differences.

In addition to the infiltration of activated T-cells into the tumor, T-cell activation in the TdLNs is critical for effective induction of cDCV-anti-tumor immunity and efficacy [28]. We also found a significant increase in CD8⁺, CD4⁺, TC1⁺ (activated T-cells), and DCs in the TdLNs from cDCV-treated mice. This was not seen in mice treated with anti-CTLA-4 alone or combination therapy. cDC1 cells are critical for cross-presenting tumor antigens to activate CD8^+^ T-cells [29]. For example, OVA-derived specific peptide SIINFEKL can be used to show activation of T-cells by cDCV cells primed by melanoma cells expressing OVA (B16-OVA cells) [14]. cDC1 cells process and present antigen peptide to activate the adaptive immune response typically by inducing T-cell gene expression such as IFN-γ, granzyme B, perforin, etc. [30]. cDC1 cells are required to generate broad CD8 responses against a range of diverse neoAgs [31]. While there is currently no *specific* osteosarcoma antigen(s) reported that can be used to rapidly generate cDCVs for patients with metastatic disease or investigate the genes that are activated in the T-cells following exposure to cDCs, we speculate that there will be induction of the IFN-g, granzyme B, and perforin genes by our cDCV. Recently, we demonstrated that CD70 is overexpressed in multiple OS cell lines and patient-derived xenografts [32,33]. We are currently investigating the use of CD70 as an OS specific antigen for activation of OS-specific cDCVs. This may provide the needed tool to investigate the mechanism of T-cell activation; we therefore expect that T-cells activated by CD70 peptide presented by cDC1 cells will show induction in IFN-γ, granzyme B, and perforin gene expression as well.

There was no significant increase in Th1, T-regs, TH17, macrophages, monocytes, or neutrophils in the TdLNs from any of the treatment groups. As we failed to show an improved immune cell profile in the tumors and TdLNs of mice treated with combination therapy, further studies are required to identify the mechanism for the enhanced therapeutic response. However, our results suggest the potential for combination therapy against OS lung metastases.

## 5. Conclusions

There have been no significant improvements in treating patients with OS metastases for almost 4 decades. Taken together, our investigations have identified a potential new immunotherapy for patients with OS, particularly those with lung metastases, which is the major cause of death in relapsed patients. Induction of a systemic response by in vitro-generated cDCV using CD103⁺cDC1 cells and OS lysates with successful infiltration of activated T-cells into the center of the tumor suggests that cDCV therapy may be able to circumvent the immunosuppressive tumor microenvironment, which has limited the activity of current immunotherapies for this cancer. The presence of increased DCs in the TdLNs following intravenous vaccine administration suggests that systemic delivery can achieve antigen presentation and T-cell activation and result in an anti-tumor T-cell-mediated response. Although we have no mechanistic explanation for this at the present time, our data also suggest that an enhanced response may be achieved by combining anti-CTLA-4 therapy with cDCV therapy. Finally, the safety and tolerability track record of DC-based therapies in children and adolescent patients supports the translational potential. In view of the poor survival rate in patients with relapsed, refractory, or metastatic OS and the absence of any effective therapies for these individuals, cDCV therapy warrants consideration for this difficult patient population.

## Figures and Tables

**Figure 1 cancers-16-03251-f001:**
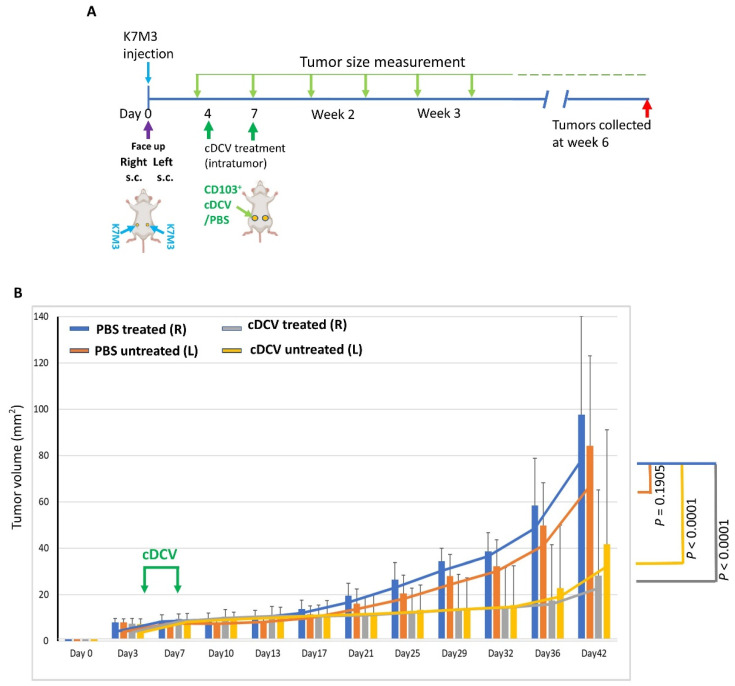
Effect of an intratumor cDCV generated from K7M3 tumor cell lysates on tumor growth and systemic immunity. (**A**) Diagram for the bilateral tumor model: 2 × 10^6^ K7M3 cells were injected subcutaneously into the right and left sides of mice. On days 4 and 7, the right-side tumors were intratumorally injected with phosphate-buffered saline (PBS, control) or the type 1 CD103^+^ dendritic cell vaccine (cDCV). The left-side tumors were untreated. (**B**) Tumor growth was assessed every 3–4 days for 5 weeks after therapy. Mice treated intratumorally with the cDCV showed a decrease in tumor growth in both the treated and untreated contralateral tumors compared to the control mice. Data were analyzed using a linear mixed effects model. L: left side; R: right side; s.c.: subcutaneous. Six to eight mice per treatment group. All values reported represent means ± SEMs.

**Figure 2 cancers-16-03251-f002:**
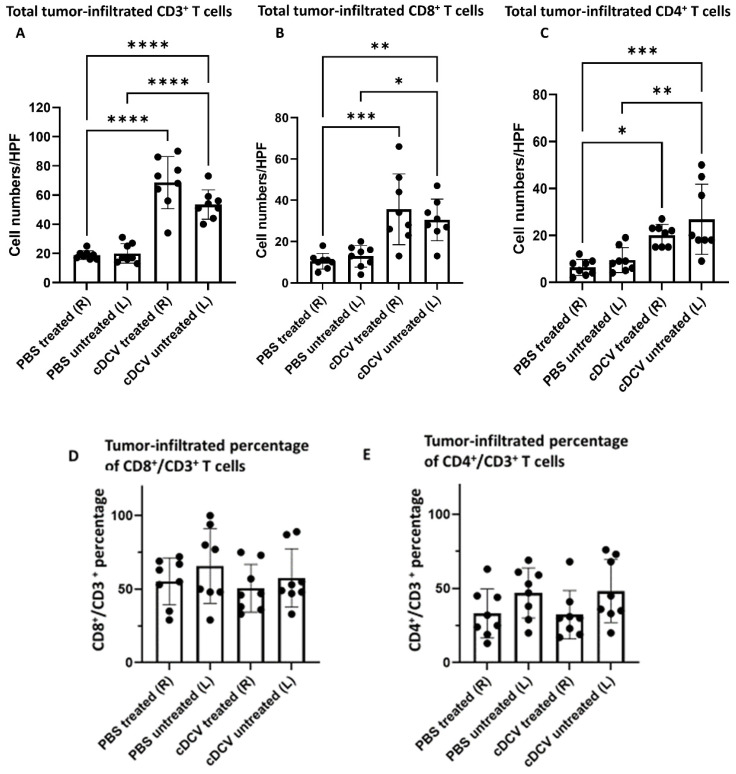
Effect of cDCV treatment on infiltrating T-cells and macrophages. Bilateral tumors were stained with anti-CD3/anti-CD8/anti-CD4 (**A**–**C**,**K**,**L**), anti-CD11c/anti-CD103 (**H**,**I**), anti-CD86 (**F**), or anti-CD163 (**G**) antibodies. Percentage of CD8^+^ or CD4^+^ cells out of CD3^+^ cells were showed in (**D**,**E**). Percentage of CD103^+^ cells out of CD11c^+^ cells was showed in (**J**). Positive cells were quantified using Simple PCI software. *n* = 7–8 mice per group. Data were analyzed using one-way ANOVA test. * *p* < 0.05, ** *p* < 0.01, *** *p* < 0.001, **** *p* < 0.0001. cDCV: type 1 CD103^+^ dendritic cell vaccine; HPF: high-power field; L: left side; PBS: phosphate-buffered saline; R: right side. All values reported represent means ± SEMs. Arrows indicates an example of a positive stained cell in each field.

**Figure 3 cancers-16-03251-f003:**
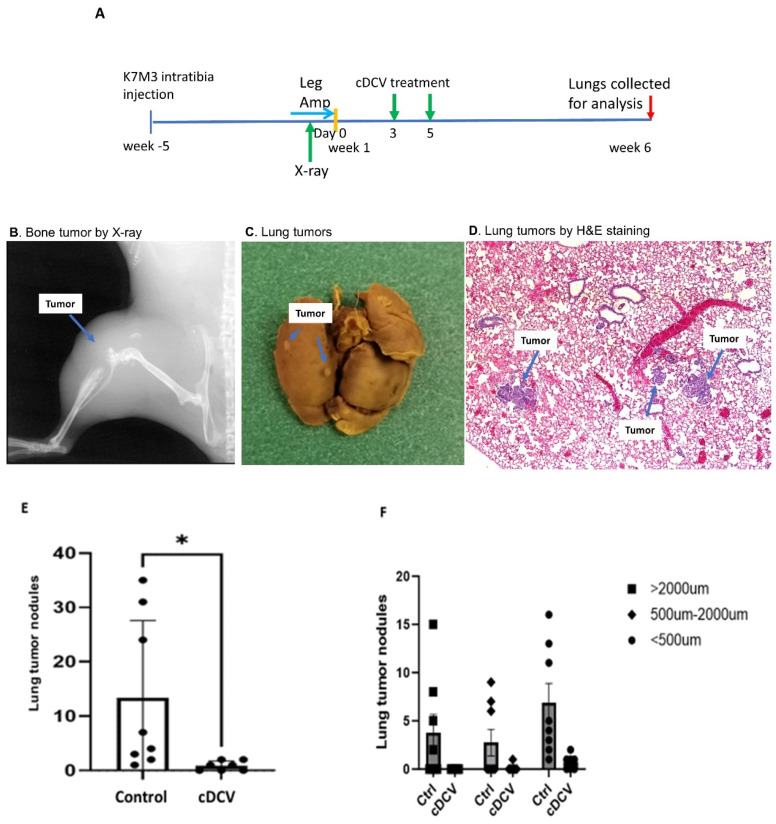
Effect of systemic cDCV therapy on established lung metastases. (**A**) Schematic diagram for systemic cDCV therapy. K7M3 cells were injected into the right tibia. Primary tumor and lung metastases were confirmed at 5 weeks using X-ray (**B**), Bouin’s Fixation solution (**C**), and hematoxylin and eosin staining (**D**). The tumor leg was then amputated. Starting 3 days after surgery, mice were treated with cDCV twice on day 3 and day 5. Lungs were collected 6 weeks after amputation. The number and sizes of tumor nodules was quantified using H&E staining (**E**,**F**); 7–8 mice per treatment group. Data were analyzed using the student *t* test. * *p* < 0.05. Amp: amputation; cDCV: type 1 CD103^+^ dendritic cell vaccine. All values reported represent means ± SEMs.

**Figure 4 cancers-16-03251-f004:**
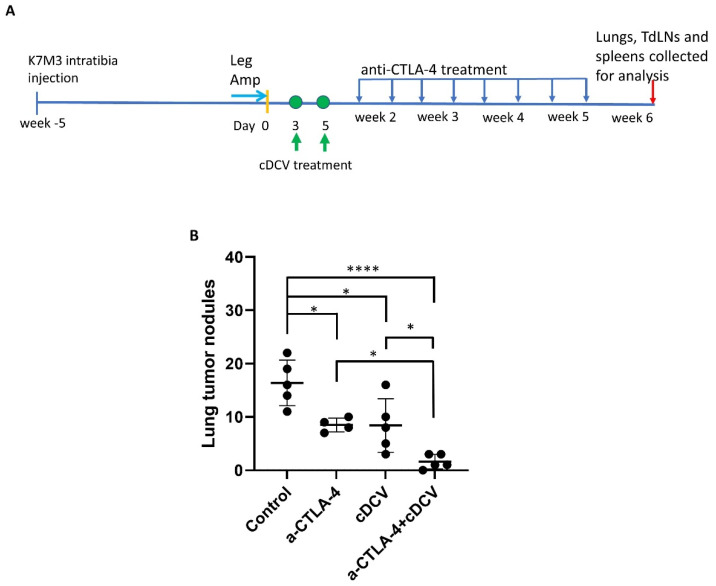
Effect of combining cDCV therapy with anti-CTLA-4 therapy. (**A**) Schematic diagram for combining cDCV therapy with anti-CTLA-4 therapy. K7M3 cells were injected into the right tibia. The tumor bearing legs were amputated at 5 weeks. Starting 3 days after surgery, mice were treated with (Control): i.v. PBS twice as control treatment on day 3 and 5; (a-CTLA-4): i.p. anti-CTLA-4 2×/week for 4 weeks; (cDCV): i.v. cDCV twice on day 3 and 5; or (a-CTLA-4+cDCV): i.v. cDCV twice plus i.p. anti-CTLA-4 2×/week for 4 weeks thereafter. Five mice per treatment group. Lungs, TdLN, and spleens were collected 6 weeks after amputation (1 week after therapy completion). (**B**) The number of tumor nodules was quantified using hematoxylin and eosin (H&E) staining. *n* = 5 mice per group. Data were analyzed using a one-way analysis of variance (ANOVA) test. * *p* < 0.05, **** *p* < 0.0001. Amp: amputation; cDCV: type 1 CD103⁺ dendritic cell vaccine; CTLA-4: cytotoxic T-lymphocyte antigen-4; TdLNs: tumor-draining lymph nodes. All values reported represent means ± SEMs.

**Figure 5 cancers-16-03251-f005:**
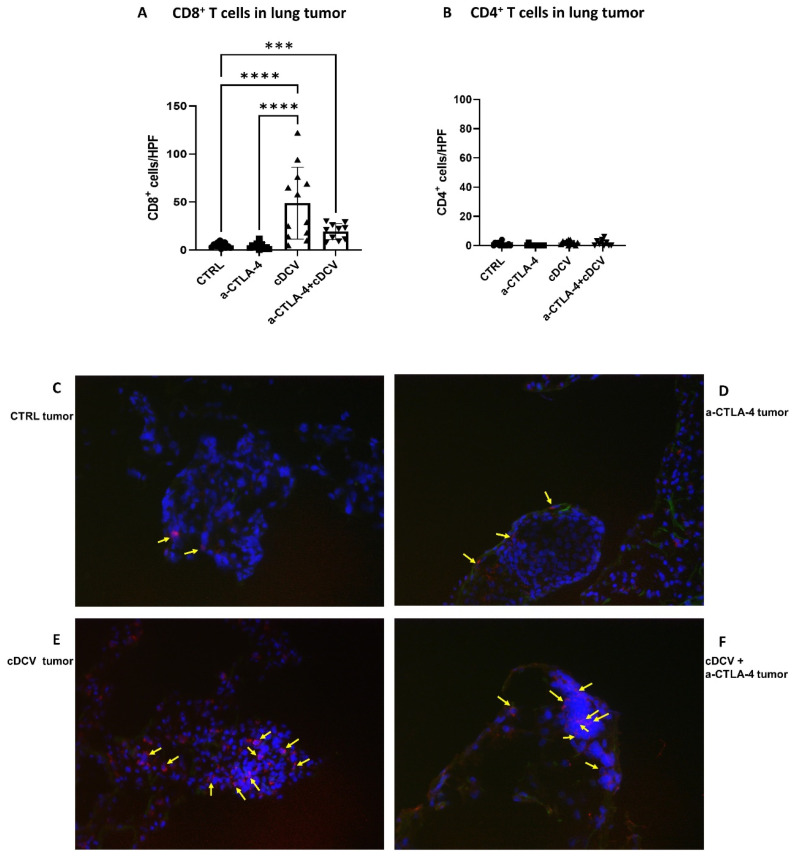
Effect of cDCV therapy with or without anti-CTLA-4 on T-cell infiltration in tumors. Tumors from the mice in Figure 4 were analyzed for the infiltration of T-cell subsets ((**A**) CD8+ T-cells and (**B**) CD4+ T-cells). Data were analyzed using Mann–Whitney U-test. *** *p* < 0.001, **** *p* < 0.0001. (**C**–**F**) Representative images of lung tumor nodules were analyzed for CD8^+^ cells using immunofluorescence staining. Arrows indicate the positive CD8^+^ stained cells in each field. cDCV: type 1 CD103⁺ dendritic cell vaccine; CTLA-4: cytotoxic T-lymphocyte antigen-4; CTRL: control; HPF: high-power field. All values reported represent means ± SEMs.

**Figure 6 cancers-16-03251-f006:**
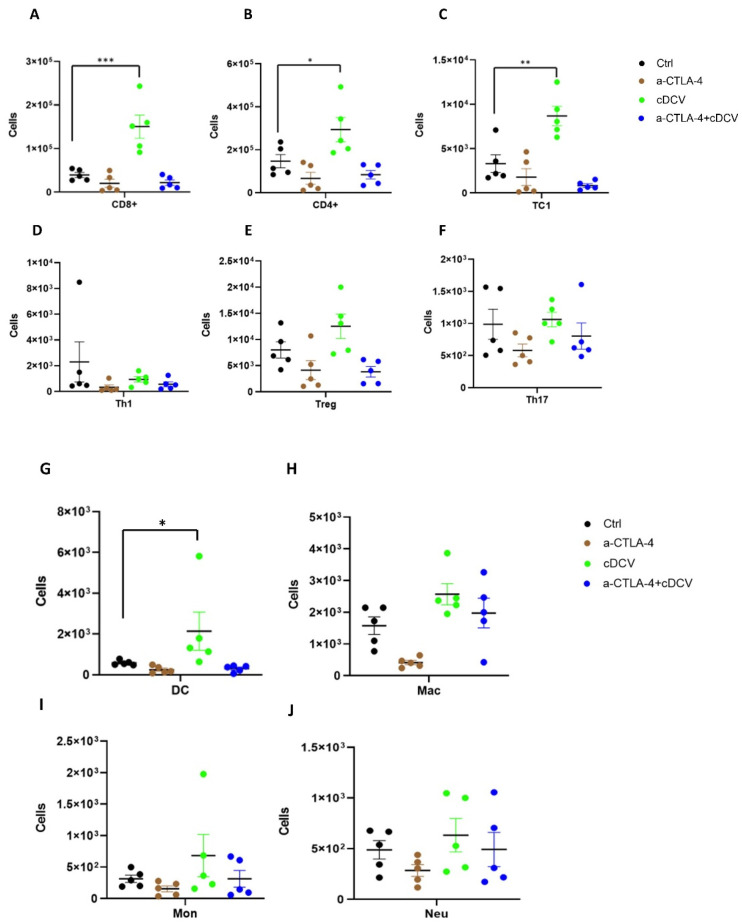
Effect of cDCV therapy on the immune cell profile of the TdLNs. The TdLNs from the mice in Figure 4 were analyzed for the T-cell subsets, and the myeloid cells using fluorescence-activated cell sorting (FACS). (**A**) CD4+ cells, (**B**) CD8+ cells, (**C**) TC1s, (**D**) Th1s, (**E**) Treg cells, (**F**) Th17 cells, (**G**) DCs, (**H**) macrophages, (**I**) monocytes, and (**J**) neutrophils. Data were analyzed using Mann–Whitney U-test. * *p* < 0.05, ** *p* < 0.01, *** *p* < 0.001. cDCV; type 1 CD103⁺ dendritic cell vaccine; CTLA-4: cytotoxic T-lymphocyte antigen-4; Ctrl: control; DCs: dendritic cells; Macs: macrophages; Mons: monocytes; Neus: neutrophils; TC1s: cytotoxic T-cells; TdLNs: tumor-draining lymph nodes; Th1s: T-helper 1 cells; Th17s: T-helper 17 cells; Tregs: regulatory T-cells. All values reported represent means ± SEMs.

## Data Availability

The data presented in this study are available on request from the corresponding author. Appendix A are available online as Appendix A.

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
