# Peer review of "CD103+ cDC1 Dendritic Cell Vaccine Therapy for Osteosarcoma Lung Metastases"

_cancers, 2024, doi:10.3390/cancers16193251_

Round 1
Reviewer 1 Report
Comments and Suggestions for Authors
This study aims to investigate the use of in vitro-generated DC103 to elicit an immune response against osteosarcoma. Although the authors demonstrated significant tumor control results from various in vivo experiments, the data analysis requires improvement for clarity and persuasiveness.
1. It appears that all data related to the quantification of immune cells infiltrating tumors were obtained solely through immunohistochemistry. Please provide detailed descriptions of how the cells were quantified, including the number of slides per sample and the number of fields examined per slide. Additionally, please include representative imaging data for Figure 2.
2. Has any imaging data been validated using flow cytometry?
3. How did your vaccine increase the number of CD8+ and CD4+ T cells within tumors? Was this increase due to enhanced migration, better expansion, or reduced cell death within the tumors?
4. Besides increasing T cell numbers, did your vaccine enhance the function of T cells infiltrating the tumors? Did you assess their ability to produce effector cytokines such as IFNγ, TNFα, and Granzyme B?
Author Response
We appreciate all the opinions, comments and suggestions from the reviewer. We have responded to the comments from both reviewers and edited the manuscript accordingly.
Point 1. It appears that all data related to the quantification of immune cells infiltrating tumors were obtained solely through immunohistochemistry. Please provide detailed descriptions of how the cells were quantified, including the number of slides per sample and the number of fields examined per slide. Additionally, please include representative imaging data for Figure 2.
Reply:
- Detailed procedure for cell quantification is updated in the method section (Lines 213-218). “Three tumor frozen samples from each treatment group were sectioned for CD3/CD4/CD8 staining and CD11c/CD103 staining. Tumor areas from each section were imaged at 400× by Leica DMi8 microscope (Leica Microsystems, Wetzlar, Germany). Positive staining was quantified in eight random microscopy fields from three tumor samples using Simple PCI software (Hamamatsu, Sewickley, PA), and the average expression was calculated as averages ± SEM.”
- Representative images were added as Figure 2K-2L of the IFC staining of CD3/CD4/CD8 for the T-cells in the tumor area. The main text and figure 2 legend has been updated accordingly.
Point 2. Has any imaging data been validated using flow cytometry?
Reply:
After collecting the final tumor samples we must chose to do either imaging on frozen samples or do single cell dissociation for flow cytometry. We can not do both on the same sample. Using the same tumor and processing half of the tumor may not be reflective of what is happening in the whole tumor and can produce inaccurate data as T-cell infiltration is most likely not uniform.
T cell infiltration in osteosarcoma tumors from cDCV vaccine-treated mice or PBS-treated controls was previously analyzed by flow cytometry after single cell suspensions were generated from the tumors (Reference 14, Figure 5C). We demonstrated increased T-cell infiltration into the cDCV-treated tumors, similar to what was report in this study using IFC staining.
Point 3. How did your vaccine increase the number of CD8+ and CD4+ T cells within tumors? Was this increase due to enhanced migration, better expansion, or reduced cell death within the tumors?
Reply:
The mechanism responsible for the increased numbers of CD8+ and CD4+ T-cells within the tumor was not investigated in the present study. As the reviewer noted the increase can be explained by enhanced migration to the tumor and penetration into the tumor, T-cell expansion within the tumor, or a decrease in T-cell death. Our experiments reported in Ref 14 showed that the cDCs generated for cDCV therapy triggered T-cell division in vitro.
Point 4. Besides increasing T cell numbers, did your vaccine enhance the function of T cells infiltrating the tumors? Did you assess their ability to produce effector cytokines such as IFNγ, TNFα, and Granzyme B?
Reply:
We previously reported that cDCV therapy resulted in an increase in the number of IFNg+ CD8 cells in the OS metastases generating using i.v. injection of OS cells (experimental metastasis model). These studies were not repeated here using the intra-bone orthotopic metastasis model.
Reviewer 2 Report
Comments and Suggestions for Authors
The authors presented effect of CD103+cDC1 dendritic cell vaccine therapy for osteosarcoma patients. This is important study that decreased metastatic development in vivo. I have several major points to address.
1. Figure 1. Indicate how many mice were used for each group. It is important to include and present data on tumor growth curves for individual mice. The standard deviations are very high for each group at day 42.
2. The rechallenge model will be good to use for vaccine effect.
3. The known peptide antigens can be used to show activation of T cells. The genes activated in single cells can show mechanism of T cell activation. This could be discussed.
4. Human clinical applications and trials with similar approaches will be important to include in detail to Discussion.
Author Response
We appreciate all the opinions, comments and suggestions from the reviewer. We have responded to the comments from both reviewers and edited the manuscript accordingly.
Point 1. Figure 1. Indicate how many mice were used for each group. It is important to include and present data on tumor growth curves for individual mice. The standard deviations are very high for each group at day 42.
Reply:
We used 6 mice for the control treatment group and 8 mice for the cDCV treatment group. Tumor growth curves for individual mice are plotted and added as Supplemental Figure 2 A-D. The Figure legend has also been added and updated. The original supplemental Figure 2 has been renamed as Supplemental Figure 3.
The high standard deviation on day 42 were due to the exponential growth of some of the subcutaneous tumors on the last week of the study. This is not unexpected as the last therapy was given on day 7. Mice went untreated for ~ 5 weeks may be allowed for tumor escape. The treatment was therefore not intended to be curative therapy. Curative therapy will most likely require multiple cDCV injections similar to what was used with other immunotherapies targeting innate immune cells i.e. L-MTP-PE (Reference 9, 10, 22). Therefore a linear mixed effects model was used to analyze all data points together. We have added text in the Discussion to recognize that the cDCV therapy as administered in this study was not curative. Please see page 15 lines 416 to 429.
Point 2. The rechallenge model will be good to use for vaccine effect.
Reply:
We performed a rechallenge experiment in our prior report (reference 14), where surviving mice who had received cDCV treatment were rechallenged with K7M3 cells, without receiving additional therapy. Tumor-free survival was observed in the majority of the mice up to 370 days following tumor rechallenge. This demonstrated that the cDCV vaccination elicited long-lasting protection against osteosarcoma. We elected not to repeat these studies here.
Point 3. The known peptide antigens can be used to show activation of T cells. The genes activated in single cells can show mechanism of T cell activation. This could be discussed.
Reply:
cDC1 is critical for cross-presenting tumor antigens to activate CD8+ T cells (29), for example, OVA-derived specific peptide SIINFEKL can be used to show activation of T-cells by cDCV cells primed by melanoma cells expressing OVA (B16-OVA cells) (14). cDC1 cells process and present antigen peptide to activate the adaptive immune response typically by inducing T-cell gene expression such as IFN-γ, Granzyme B, perforin etc. (30). cDC1 are required to generate broad CD8 responses against a range of diverse neoAgs (31). While there is currently no specific osteosarcoma antigen(s) reported that can be used to rapidly generate cDCVs for patients with metastatic disease or investigate the genes that are activated in the T-cells following exposure to cDCs, we speculate that there will be induction of the IFN-g, Granzyme B and perforin genes by our cDCV. Recently we demonstrated that CD70 is overexpressed in multiple OS cell lines and patient-derived xenografts (32, 33). We are currently investigating the use of CD70 as an OS-specific antigen for activation of OS-specific cDCVs. This may provide the needed tool to investigate the mechanism of T-cell activation and the genes that are activated. We therefore expect that T-cell activated by CD70 peptide presented by cDC1 cells will show induction of IFN-γ, Granzyme B and perforin gene expression as well. We have added this to the discussion section as requested at page 17 Line 527-542.
Point 4. Human clinical applications and trials with similar approaches will be important to include in detail to “Discussion”.
Reply:
This has been added to the discussion, please see page 15 lines 416-429. Based on our studies, we hypothesize that cDCV therapy for OS needs to be administered for a prolonged period of time similar to the trials done with the macrophage activating agent L-MTP-PE (9, 10, 22). When administered weekly for 6 months both the disease-free and overall survival were significantly increased in patients with OS lung metastases (9, 22). As we showed with L-MTP-PE, we also anticipate the cDCV therapy can be administered together with the chemotherapy agents used to treat OS (10, 23, 24).
Round 2
Reviewer 1 Report
Comments and Suggestions for Authors
The authors have addressed my questions successfully.